# *SCAPER*-Related Autosomal Recessive Retinitis Pigmentosa with Intellectual Disability: Confirming and Extending the Phenotypic Spectrum and Bioinformatics Analyses

**DOI:** 10.3390/genes15060791

**Published:** 2024-06-16

**Authors:** Rajech Sharkia, Abdelnaser Zalan, Amit Kessel, Wasif Al-Shareef, Hazar Zahalka, Holger Hengel, Ludger Schöls, Abdussalam Azem, Muhammad Mahajnah

**Affiliations:** 1Unit of Human Biology and Genetics, The Triangle Regional Research and Development Center, Kafr Qara 3007500, Israel; dr.zalan@hotmail.com (A.Z.); w.shareef@gmail.com (W.A.-S.); 2Unit of Natural Sciences, Beit-Berl Academic College, Beit-Berl 4490500, Israel; 3Baqa College, Al-Qasmi Street, 64, Baqa Al-Gharbia 3010000, Israel; 4Department of Biochemistry and Molecular Biology, Faculty of Life Sciences, Tel-Aviv University, Tel-Aviv 69978, Israel; amitkessel@tauex.tau.ac.il (A.K.); azema@tauex.tau.ac.il (A.A.); 5Child Development and Pediatric Neurology Service, Meuhedet—Northern Region, Hadera 38100, Israel; hazarzdr@gmail.com; 6Department of Neurology and Hertie-Institute for Clinical Brain Research, University of Tübingen, 72074 Tübingen, Germany; holger.hengel@uni-tuebingen.de (H.H.); ludger.schoels@uni-tuebingen.de (L.S.); 7German Center of Neurodegenerative Diseases (DZNE), 72076 Tübingen, Germany; 8The Ruth and Bruce Rappaport Faculty of Medicine, Technion-Israel Institute of Technology, Haifa 31096, Israel; muhamadmah@hymc.gov.il; 9Child Neurology and Development Center, Hillel Yaffe Medical Center, Hadera 38100, Israel

**Keywords:** *SCAPER*, intellectual disability, retinitis pigmentosa, Arab society in Israel

## Abstract

Mutations in the gene *SCAPER* (S phase Cyclin A-Associated Protein residing in the Endoplasmic Reticulum) have recently been associated with retinitis pigmentosa (RP) and intellectual disability (ID). In 2011, a possible involvement of *SCAPER* in human diseases was discovered for the first time due to the identification of a homozygous mutation causing ID in an Iranian family. Later, five studies were published in 2019 that described patients with autosomal recessive syndromic retinitis pigmentosa (arRP) accompanied by ID and attention-deficit/hyperactivity disorder (ADHD). This present study describes three patients from an Arab consanguineous family in Israel with similar clinical features of the SCAPER syndrome. In addition, new manifestations of ocular symptoms, nystagmus, glaucoma, and elevator palsy, were observed. Genetic testing of the patients and both parents via whole-exome sequencing revealed the homozygous mutation c.2023–2A>G in *SCAPER*. Phenotypic and genotypic descriptions for all available cases described in the literature including our current three cases (37 cases) were carried out, in addition to a bioinformatics analysis for all the genetic variants that was undertaken. Our study confirms and extends the clinical manifestations of SCAPER-related disorders.

## 1. Introduction

In chromosome 15q24.3, the *SCAPER* gene (S phase Cyclin A-Associated Protein residing in the Endoplasmic Reticulum; OMIM#611611) encodes for a 158 kDa SCAPER protein, which interacts with cyclin A and functions as a feedback loop regulator in the G1/S and G2/M phases of the cell cycle [1]. Some researchers (Najmabadi and coworkers) [2] first proposed a possible involvement of the *SCAPER* gene in human disease by identifying a homozygous frameshift *SCAPER* variant as the cause of non-syndromic intellectual disability (ID) in a family from Iran. A patient with biallelic loss of function (LOF) *SCAPER* mutations linked to retinal illness was later described by Carss and colleagues [3]. Since then, seven individuals from five families in Spain, Israel, and Iran have had biallelic LOF variants linked to ID with or without retinitis pigmentosa (RP) [4,5].

More recently, eight individuals from two consanguineous Bedouin families belonging to the same tribe in southern Israel were found to have a *SCAPER* gene variant. This variant is associated with a presentation resembling Bardet-Biedl syndrome (BBS), which includes ID, RP, short stature, obesity, and brachydactyly. Additionally, preliminary functional studies suggested that SCAPER may play a role in ciliary dynamics and disassembly [6].

Later on, it was found that syndromic autosomal recessive retinitis pigmentosa (arRP) with ID and attention-deficit/hyperactivity disorder (ADHD) [7] as well as non-syndromic arRP [8] have both been linked to pathogenic variations in the *SCAPER* gene. Moreover, Fasham and coworkers describes clinical and genetic findings of six patients of Amish, Caucasian, and South Asian descent. They carried out a comprehensive phenotypic comparison in relation to the previously reported 17 cases with *SCAPER* variants [9]. This analysis revealed the presence of a variable pattern of dysmorphic features associated with *SCAPER* gene variants. In the latest study [10], the researchers expanded the phenotypic spectrum of *SCAPER* pathogenic variants associated with a relatively milder form of syndromic RP compared to what has been previously reported [9].

Our current study includes a full clinical description of three patients from an Arab consanguineous family in Israel who presented similar clinical features of the SCAPER syndrome, with the extension of new ocular symptoms: nystagmus, glaucoma, and elevator palsy. We present a comprehensive clinical and genetic description in comparison with the previously reported 34 cases. Additionally, a bioinformatics analysis of the mutational variants of SCAPER protein was carried out.

## 2. Materials and Methods

In this current study, all available cases with *SCAPER* gene variants (until the date of preparation of this study, mid-year 2024) were enlisted with their clinical features (Appendix A). The literature review was carried out extensively using the PubMed and Google Scholar websites (https://pubmed.ncbi.nlm.nih.gov/, accessed on 10 February 2023 and https://scholar.google.com/, accessed on 15 March 2024). The relative frequencies of various clinical features in all these cases were summarized (Table 1). Additionally, all the *SCAPER* gene variants identified in patients with ID and RP were presented (Table 2).

### 2.1. Genetic Analysis

DNA was extracted from peripheral blood lymphocytes of all family members and whole exome sequencing (WES) was performed as previously described [11]. The genetic testing via WES revealed a pathogenic homozygous splice variant c.2023-2A>G in *SCAPER* (reference chromosome and transcript sequences are from NC_000015.10 and NM_020843.4, respectively). This genetic splice variant was first described as pathogenic in 2017 [5]. Sanger sequencing of all family members showed full segregation with the disease. The parents and the healthy brother were heterozygous for the variant.

### 2.2. Structural Analysis of the Mutational Effects

#### 2.2.1. Structure Prediction of Human SCAPER

The three-dimensional structure of SCAPER was predicted by the AlphaFold2 software [12]. The calculation relied on the sequence of human SCAPER (UniProt ID: Q9BY12).

#### 2.2.2. Prediction of SCAPER’s Binding Regions

The binding of SCAPER to human cyclin A (UniProt ID Q9BY12), CDK2 (UniProt ID P20248), and the CNOT11 subunit of the CCR4-NOT complex (UniProt ID Q9UKZ1) was predicted using AlphaFold-Multimer software [13] with energy minimization.

#### 2.2.3. Prediction of SCAPER’s Binding Affinity Change Following Mutations

The prediction was carried out using the mCSM web server [14,15]. For the calculation, we used the AlphaFold-Multimer models of SCAPER’s complexes with cyclin A, CDK2, CNOT11, and the server’s default parameters.

#### 2.2.4. Prediction of SCAPER’s Thermodynamic Stability Change Following the F628C Mutation

The prediction was carried out using the DUET web server [16] with default parameters and the AlphaFold2 model of SCAPER.

#### 2.2.5. Calculation of SCAPER’s Evolutionary Conservation

The prediction was carried out using the ConSurf web server [17,18], with 282 homologues that were found in the UNIREF-90 database by HMMER [19,20] and aligned by MAFFT [21]. The calculation was performed using the empirical Bayesian algorithm [22].

## 3. Results

### 3.1. Case Presentation

This research was approved by the ethics committee of the Hillel Yaffe Medical Center in the city of Hadera in Israel. A family from a village of Arab society in Israel was referred to the pediatric neurology clinic of this medical center. The parents were first-cousin relatives with three affected daughters and a healthy son. The affected daughters suffered from global developmental delay, ID, and RP. The detailed clinical manifestations of these patients are described in Table 3.

#### 3.1.1. Patient A

The eldest of the three sisters was a 30-year-old woman who was born after normal pregnancy and delivery. Her weight at birth was normal (2700 g). She had mild motor developmental delay, walked at 15 months, spoke her first words at 12 months, and had slow acquisition of language milestones. At the age of two, she was diagnosed with language delay and started speech therapy. At the age of 6 years, she was diagnosed with ADHD and treated with methylphenidate. Intelligence tests were undertaken at the age of 12 years and showed intermediate ID with an IQ of 52. A brain MRI scan showed nonspecific findings (a high-intensity signal in T2 at the left white periventricular matter). The auditory examination was normal. Ophthalmological examination revealed RP and strabismus. She underwent a strabismus repair surgery and later on was diagnosed with left eye glaucoma and cataract. In the last examination at the age of 30 years, her height was 147 cm, her weight was 71.9 kg, she had a BMI = 33.3 (obese), and her OFC was 53.5 cm (percentile 20). She has facial dysmorphic features (prominent nose, thin upper lip, thick eyebrows, round face, and high forehead). She also had mild left nystagmus, nasopharyngeal insufficiency, and severe muscle hypotonia, including fascial musculature.

#### 3.1.2. Patient B

The younger affected sister, 29 years old, was born after a normal pregnancy and delivery. She was diagnosed with mild developmental motor delay and moderate language delay. She started to walk at the age of 16 months and to speak after 2 years of age. At 3 years of age, she was diagnosed with RP, left eye strabismus (exotropia), and nystagmus, and later on, she also suffered from glaucoma and left superior rectus palsy (double elevator palsy). She was also diagnosed with intermediate ID (IQ 55) and ADHD at 7 years old. In the last examination at the age of 29 years, her height was 153 cm, her weight was 50 kg, she had a BMI of 21.4 (normal), and her OFC was 53.5 cm (percentile 20). She has facial dysmorphic features (prominent nose, thin upper lip, thick eyebrows, round face, high forehead), prominent hypotonia, and nasopharyngeal insufficiency.

#### 3.1.3. Patient C

The youngest affected sister, 14 years old, was born after normal pregnancy and delivery. She was diagnosed with mild developmental motor and speech delay and started walking at 15 months and speaking at 18 months. At the age of 7, her cognitive tests revealed intermediate ID (IQ 59) and ADHD. Her ophthalmological examination revealed RP and strabismus.

Recently, at 14 years of age, her height is 152 cm, her weight is 86 kg, she has a BMI of 37.2 (obese), and her OFC is 56 cm (percentile 75). She has facial dysmorphic features similar to her sisters (prominent nose, thin upper lip, thick eyebrows, round face, high forehead), nasopharyngeal insufficiency, and muscle hypotonia.

### 3.2. Phenotypic Features and Their Relative Frequencies

Since the first case of the association of the *SCAPER* gene with ID that was described in 2011 [2], to date, about 37 cases have been diagnosed with the implication of the *SCAPER* gene syndrome. In this current study, a review of all these cases along with their clinical manifestations was carried out and is presented in Appendix A. Further, the relative frequencies of various clinical features of all the cases were calculated and are shown in Table 1.

It was found that all the patients with SCAPER syndrome that had been reported, including our cases, were affected with ID, except for a single case described by Jauregui and coworkers [8]. Additionally, about 75% of the patients (21 out of 28) had behavioral disorders, where ADHD was the predominant one accounting for 52% of the cases (13 out of 25), whereas self-injury accounted for ~13% of the cases (two out of fifteen). Autism, dyspraxia, and skin-picking behavior were reported in one patient each. It was also noted that seizures were reported in only one study (three patients), as reported by Kahrizi and coworkers [7]. Furthermore, developmental delays (including sitting, standing, walking, and speaking) were considered to be evident symptoms of the disease; however, normal MRI was revealed in the majority (~89%) of the patients.

Ocular involvement is also considered to be a key clinical characteristic of the SCAPER syndrome. The main symptom was RP, which was present in 31 cases of the total 33 reported patients (including our three patients). In about 77% of all cases for whom the data are clearly mentioned, reduced and progressive loss of night vision was observed. Less common ocular symptoms were myopia, strabismus, and cataracts, which had frequencies of 56%, 56%, and 37% of the cases, respectively. In addition to all these ocular symptoms, two of our three patients had nystagmus and glaucoma, while one of them also had elevator palsy.

Despite the fact that other clinical characteristics, including facial dysmorphism (~32%), skeletal abnormalities (~27%), and hypotonia (~27%), were less common, our patients presented all of these dysmorphic features. It was found that the majority of the patients’ BMIs were out of the normal range (about 74%), as ~53% of the patients were obese and overweight while 21% of the patients were underweight.

Additionally, it was observed that consanguinity was associated with the majority of the cases (~90%) of the *SCAPER* gene disease.

### 3.3. Genotypic Variations

The *SCAPER* gene variants identified in patients with ID and RP were collected and presented in Table 2. The results revealed that the majority of the *SCAPER* gene variants were nonsense mutations, as they were observed in 27 cases (~73% of all the cases). Three splice site variants were observed: the first is c.2023–2A>G, which was detected in six cases; the other two splice site variants, c.1495+1G>A and c.2166-3C>G, were present in two different patients [9].

There are two missense mutations (S1219N and F628C), which both had about 3% relative frequency each, and each of them was detected in only one separate case. The SCAPER variants discovered so far are recessive disease variants. These variants are either absent from gnomAD (v4.1.1) or present at a lower frequency than expected (allele frequencies below 0.001%; see Table 2).

### 3.4. Bioinformatic Analysis of the SCAPER Gene Product

Predicting the molecular basis of mutations requires knowledge of the three-dimensional structure and functional regions of the protein in which they appear. Therefore, before we analyze the mutations, we explore SCAPER’s structure–function relationships, as described below.

#### 3.4.1. SCAPER’s Structure and Known Functional Domains

Although poorly characterized, SCAPER is known to contain a few functional domains and motifs (see highlighted parts in the AlphaFold model of the protein in Figure 1) [1,23]. Two domains that are normally associated with SCAPER are the SCAPER_N (positions 90–183) and the zinc finger (positions 790–822) domains. SCAPER_N and the adjacent *RXL* motif (R_199_S_200_L_201_ in SCAPER) are implicated in the principal role of SCAPER, i.e., binding to the cell cycle factor cyclin A2 when bound to the kinase CDK2, which allows SCAPER to regulate cellular reproduction [1]. The zinc finger domain of SCAPER is of the C2H2-type, which normally appears in transcription factors [24] and may allow SCAPER to interact with the cell’s genome or RNA molecules. The middle part of SCAPER contains a long helix, part of which is arranged as a coiled coil. Towards its end, SCAPER contains an α-solenoid, C-terminal domain (CTD; positions 859–1357). Recently, Hegde and co-workers implicated the CTD as a part of another key cellular process—the autoregulation of tubulin in cells [25]. By using cryo-EM, the authors demonstrated that the tubulin-specific ribosome-binding factor TTC5 recruits SCAPER to the ribosome (Figure 2). In turn, SCAPER binds and activates the CCR4-NOT de-adenylase complex, which starts the degradation process of tubulin’s mRNA during its translation by the ribosome. The cryo-EM structure of the complex shows that SCAPER forms contacts with TTC5 through positions 1302, 1305, 1338–1340, and 1343 (yellow spheres in Figure 2). SCAPER also forms contacts with the ribosomal protein uL23 through positions 930, 933–934, 937, and 941 (Figure 2, blue spheres), and with the ribosomal RNA through positions 907 and 910. Tubulin autoregulation determines the concentrations of free tubulin, which, in turn, affects the dynamics of microtubule polymerization. Thus, interfering with this process is expected to have pronounced effects on the structure of microtubules in the cell, resulting, among other things, in chromosomal segregation errors during mitosis. The nervous system is known to be highly sensitive to such mitotic defects [26,27].

#### 3.4.2. Prediction of Additional Functional Regions with AlphaFold-Multimer

Since many of the mutations analyzed in this study involve truncations that lead to the loss of entire domains in SCAPER, we attempted to obtain more information about the domains. Specifically, we focused on parts of SCAPER that might participate in interactions with known binding partners of the protein. AlphaFold2 is a machine-learning tool that can predict the three-dimensional structures of proteins with very high accuracy, based on the co-evolution of their residues [12]. A recent implementation of this tool, dubbed AlphaFold-Multimer, can be used to predict the interactions of two or more protein chains, as well as the structures of their complexes [13]. We therefore used this tool to predict the interactions of SCAPER with known binding partners. Since large parts of SCAPER are loops, which may be inherently disordered, we did not focus on the actual structures of SCAPER’s complexes with its binding partners but rather on identifying its interacting parts.

We started with the part of SCAPER that contains residues 184–213. As mentioned above, residues R_199_S_200_L_201_ that are included in this part are known to interact with CDK2-bound cyclin A [1]. Our AlphaFold-Multimer calculations suggested that the region in SCAPER that binds to human cyclin A2 (UniProt ID P20248) is larger than the above three-residue segment and it includes residues 196–203 (Figure 3). Evolutionary conservation calculation that we carried out using ConSurf software [17,18] shows that within this region, positions 198–199 are conserved and positions 201–203 are highly conserved. The next part of SCAPER we searched for is the one interacting with CDK2. Because we had no indication of where this part may be located within SCAPER, we carried out the AlphaFold-Multimer calculations with multiple segments of the protein. Our results predicted that SCAPER’s segment, which binds human CDK2 (UniProt ID P24941), is located at positions 490–498. Within this region, positions 490, 495, and 499 are evolutionarily conserved, suggesting that they form direct contacts with CDK2. The putative CDK2-binding region is distant from the cyclin A-binding region in the AlphaFold model of SCAPER (Figure 1). This probably results from the fact that the cyclin A and CDK2-binding regions of SCAPER are located on loops, whose three-dimensional structures are poorly predicted by AlphaFold. In addition, the loops probably undergo significant conformational changes upon binding to the cyclin A-CDK2 complex, a known phenomenon in intrinsically unstructured protein segments.

In the study by Hedge and co-workers, the authors used AlphaFold-Multimer to characterize the interaction between SCAPER and the CNOT11 component of CCR4-NOT [25]. Their calculations suggested that residues E620, F628, I629, and L632 in the long helix of SCAPER are the binding sites of the CNOT11. Our AlphaFold-Multimer calculations with human CNOT11 (UniProt ID Q9UKZ1) support these results, yet they extend SCAPER’s CNOT11-binding region to position 640. Indeed, this part of SCAPER is highly conserved, with 90% of the positions having the maximal ConSurf score. The interaction between this part of SCAPER and CNOT11 includes multiple hydrophobic contacts, hydrogen bonds involving E625, H639, and D640, and salt bridges involving E625 and D640 (Figure 4).

#### 3.4.3. Analysis of Mutations

##### Early Termination Mutations

Most of the mutations covered here lead to early termination of the protein, which, in turn, leads to deletion of whole domains and functional regions or motifs. At the very least, these mutations should lead to the loss of the deleted domain’s functions. Indeed, the aforementioned study of Hedge and co-workers found that the pathogenic termination mutations at positions 726 and 935, both of which lead to the deletion of SCAPER’s CTD, abolish the tubulin autoregulation process. Another possible outcome of early termination is preventing the expression or folding of the protein altogether. Based on the domain characterization given in the previous sections and on the assumption that loss of C-terminal domains does not prevent the expression and folding of the preceding domains, we predict the following effects of the termination mutations, from the *N*- to *C*-terminus (in the names below, Fs*(+x) designates a frameshift that leads to termination, x positions after the place of the frameshift):(A)R120*: This mutation leads to the loss of most of the protein, including all the domains described above. The loss of the cyclin A and CDK2-binding domains (and possibly also the zinc domain) is expected to disrupt SCAPER’s ability to carry out its cell cycle-regulating roles. The loss of the CTD and CNOT11-binding domains is expected to disrupt SCAPER’s ability to participate in tubulin autoregulation, which, in turn, should interfere with correct chromosomal segregation during cellular reproduction. Such mutation should have severe clinical outcomes and it is probably the most damaging of all other mutations analyzed here.(B)R277*, E290S-fs*(+7), V365C-fs*(+5), R366*, and V373S-fs-*(+21): These mutations lead to the loss of the CDK2 and CNOT11-binding domains, as well as the zinc finger and CTD. The loss of the CDK2-binding domain and the zinc finger is expected to interfere with SCAPER’s regulation of the cell cycle, although this effect may be milder, as the cyclin A-binding domain will still be intact. The loss of the CTD and the CNOT11-binding domain is expected to have similar effects to R120* on tubulin autoregulation.(C)V629R-fs*(+29) and R727*: These mutations leave the cyclin A and CDK2-binding domains intact, but lead to the loss of the CTD and the CNOT11-binding domain, as well as the zinc finger. Thus, the main effect of the mutations should be on the tubulin autoregulation process.(D)I746N-fs*(+6) and Q793*: These mutations lead to the loss of the zinc finger and the CTD, which is expected to affect mainly the tubulin autoregulation process.(E)L936*, I991fs*, P1075Q-fs*(+11), S1236T-fs*(+28), and V1261S-fs*(+26): These mutations will lead to the complete or partial loss of the CTD domain, which is expected to affect tubulin autoregulation.(F)Y118fs and V623fs: It is unclear if these frameshifts involve early termination, but even if they do not, the sequences downstream to the point of frameshift are expected to code for inactive proteins. Therefore, Y118fs is expected to have similar effects to the R120* mutation, and V623fs is expected to have similar effects to the V629R-fs*(+29) mutation, and both are described above.

##### Point Mutations


(A)F628C: F628 is suggested by the AlphaFold-Multimer calculations of Hegde and workers, as well as by our calculations, to be part of SCAPER’s CNOT11-interacting domain. Our calculations further suggest that this residue forms extensive van der Waals and hydrophobic interactions with residues in two helices of CNOT11, which are packed against it in the complex (Figure 4). Thus, the replacement of F628 with the smaller and more polar cysteine is expected to involve the loss of at least some of these interactions, thus destabilizing the complex. This is supported by our calculations with mCSM, a machine learning-based computational tool that calculates changes in binding energies following mutations [14,15]. Given the SCAPER-CNOT11 complex shown in Figure 4, mCSM estimates that the F628C mutation weakens the binding between the two proteins by about 2 kcal/mol. Moreover, the AlphaFold-predicted model of SCAPER suggests that F628 forms stabilizing van der Waals and hydrophobic interactions also with residues within SCAPER itself. Thus, in addition to destabilizing the complex with CNOT11, the F628C mutation might work by destabilizing SCAPER. This is supported by our calculations with DUET, a machine learning-based tool that predicts mutation-induced stability changes within a single chain [16]. The calculation estimates that the F628C mutation will destabilize SCAPER by 0.4 kcal/mol. Again, this should act additionally to the change in SCAPER–CNOT11 binding affinity.(B)S1219N: S1219 is buried inside the core of the CTD, where it is tightly packed against other residues (Figure 5). Therefore, replacing the serine with the larger asparagine is expected to cause steric hindrance, resulting in destabilization of the CTD. Since CTD is involved in tubulin autoregulation, this process is expected to be affected too. Indeed, Hegde and co-workers found this mutation to lower the expression levels of SCAPER (probably due to the destabilization), which, in turn, interferes with tubulin autoregulation [25].


##### Deletion Mutations


(A)ΔE620: This mutation is located at the beginning of the region in SCAPER, predicted by AlphaFold-Multimer to bind the CNOT11 subunit of CCR4-NOT. E620 does not interact with CNOT11 directly, and the study of Hegde and co-workers suggests that the deletion acts by changing the register of the α-helix containing it [25]. This change is likely to result in different positioning of the residues on this helix, which do interact directly with CNOT11 (Figure 4), thus weakening or disrupting the interaction between the two proteins.(B)ΔE675-K677: The three residues are located in SCAPER’s central α-helical domain, and they do not seem to interact with any other part of the protein in the AlphaFold-predicted model. Hegde and co-workers found this mutation to interfere with tubulin autoregulation [25], suggesting that it might affect the CTD and/or the CNOT11-binding domain. AlphaFold’s predicted aligned error (PAE) metric indicates that the spatial positioning of the long helix and the CTD with respect to each other might be different than in the model. Thus, the part containing the three deleted residues might interact with the CTD after all and affect its function. Testing this mechanism requires experimental work. Since residues E675-K677 are both solvent-exposed (based on the AlphaFold model) and highly evolutionarily conserved, it is also possible that they interact with other cellular macromolecules, and the mutation interferes with this binding.


## 4. Discussion

In this present study, we performed a comprehensive phenotypic and genotypic characterization of the 37 cases of *SCAPER* variants described in the literature so far, including our newly identified three cases. In the obtained results, it was found that the ID and the RP are the distinguishing phenotypes of SCAPER syndrome.

In relation to the distinguishing feature of ID, it was found that other neurological symptoms—behavioral disorders and developmental delays—were also considered to be common manifestations of this disorder. As it was previously reported that ADHD was the predominant one of the behavioral disorders [7,9], the presented data confirmed these findings. The other behavioral disorders (autism, seizure, dyspraxia, and skin-picking behavior) were less commonly observed. The developmental delays were observed to be evident symptoms of the disease and seemed to be global during childhood, including sitting, standing, walking, and speech delays.

The involvement of ophthalmological manifestations is also considered to be another core feature of the SCAPER syndrome. Our findings demonstrated that RP is the most common and is the distinguishing one. It is noteworthy that reduced and progressive loss of night vision was found to be highly associated with this disorder, while other ocular features—myopia, strabismus, and cataracts—had less involvement with the syndrome. The clinical visual observations in our new three cases revealed an expansion of the ophthalmological manifestations of the SCAPER disorder, as nystagmus, glaucoma, and elevator palsy were first reported in such patients.

Despite the fact that other clinical characteristics, including facial dysmorphism, skeletal abnormalities, and hypotonia, were less common (about one-third of the patients), our patients presented all of these dysmorphic features. It was found that the majority of the patients’ BMIs were out of the normal range. Based on these findings, as well as those described by Wormser and coworkers [6], obesity and overweight were considered to be a characteristic phenotype of the disease. It is uncertain if obesity is a component of the condition or if it results from metabolic issues like diminished brown adipose tissue activity and dysfunctional adipose tissue lipolysis, or from behavioral variables like excessive consumption of calories or insufficient exercise. Therefore, future investigations to explore this comorbidity are encouraged.

The SCAPER protein contains a single C2H2-type zinc finger, an RXL motif at the N-terminus, four coiled coil domains, a putative transmembrane domain, and an ER retrieval signal at the C-terminus. SCAPER protein interacts with cyclin A and functions as a feedback loop regulator in the G1/S and G2/M phases of the cell cycle [1]. Recently, it was demonstrated that its C-terminal domain (CTD) was involved in the autoregulation of tubulin in cells as another key cellular process [25]. To explore the molecular basis for the mutations in the *SCAPER* gene discussed here, we used various computational tools, which employ structural, biophysical, and evolutionary calculations. The calculations, most of which are based on state-of-the-art machine learning methods, allowed us to study the mutations in the context of SCAPER’s three-dimensional structure, thermodynamic stability, and binding to other cellular proteins. The results obtained in our study are in concordance with the existing knowledge of SCAPER and the pathogenic mutations associated with it, and with the findings of Hegde and co-workers’ recent cryo-EM study [25].

Most of the mutations we analyzed lead to early termination of the protein at various points. Therefore, such mutations are likely to act either by preventing the entire protein from expression or from folding, or by eliminating functional domains. SCAPER is poorly characterized and only two types of functions have been associated with its domains so far. One is the binding to cyclin A and CDK2, suggesting its involvement in cell cycle regulation [1]. The other is binding to ribosomal proteins and TTC5 as part of tubulin’s autoregulation process [25]. Thus, the early termination mutations analyzed here, and which involve the loss of whole domains, are expected to act at least by interfering with these functions.

It was found that there are two missense variants (S1219N and F628C), and the first one in which transition took place in the amino acid serine is known to be moderately conserved and predicted to be pathogenic by different prediction tools [5]. On the other hand, in the second variant, the transition took place in the amino acid phenylalanine, which is predicted to be damaged by bioinformatics predictions and is absent and/or observed rarely in control databases; thus, it was disease-causing [7].

The abovementioned missense variants as well as the deletion ones turn out to negatively affect the functionality of the SCAPER protein. For example, the S1219N mutation seems to act via the thermodynamic destabilization of a domain fold, whereas the F628C and ΔE620 mutations are likely to interfere with SCAPER’s binding to CCR4–NOT; F628C acts directly by weakening the binding through the loss of non-covalent interactions, whereas ΔE620 seems to act indirectly by changing the register of the interacting helix. It was noticeable that the mutation F628C, a missense one, has milder clinical symptoms. This is presented by a patient who was affected with this mutation as having an intermediate ID (IQ = 60), and had RP only, without other ocular complications [7].

The splice site variant (c.2023–2A>G) is a single base transition positioned in the conserved acceptor splice-site of intron 18, which is predicted to eliminate the intron 18 acceptor site. This mis-splicing is expected to lead to an in-frame deletion of three amino acids (E675-K677) from the SCAPER protein [5]. Based on a bioinformatics analysis, these amino acids are highly conserved and functionally important. We suggest that this variant has a common founder effect in the population of our region, as it is present in six cases that belong to the Arab ethnicity; five of them are Palestinian Arabs who live in Israel, while the sixth one is of Jordanian origin. This suggestion is also supported by the finding that this variant is absent from gnomAD, ExAC, and the NHLBI Exome variant server, as it is not a common benign variant present in the populations represented in those databases. Future discoveries that reveal further details about SCAPER’s structure-function relationships will no doubt provide a better understanding of the mechanisms underlying other pathogenic mutations, as well as known mutations, such as ΔE675-K677, whose mode of action is not entirely clear.

## 5. Conclusions

Our study confirmed and extended the clinical features of the *SCAPER* gene disease. Overall, ID and RP are the main common phenotypes of the SCAPER syndrome, other symptoms are closely associated with them. As it involves the neurological system, this led to a severe and large spectrum of variable symptoms of this disease. As SCAPER protein plays a vital role in various cellular functional processes, i.e., as a feedback loop regulator and autoregulation of tubulin, it is very important to carry out further future molecular studies in order to understand the pathophysiology of RP and ID, which may have an essential role in the process of treatment development.

## Figures and Tables

**Figure 1 genes-15-00791-f001:**
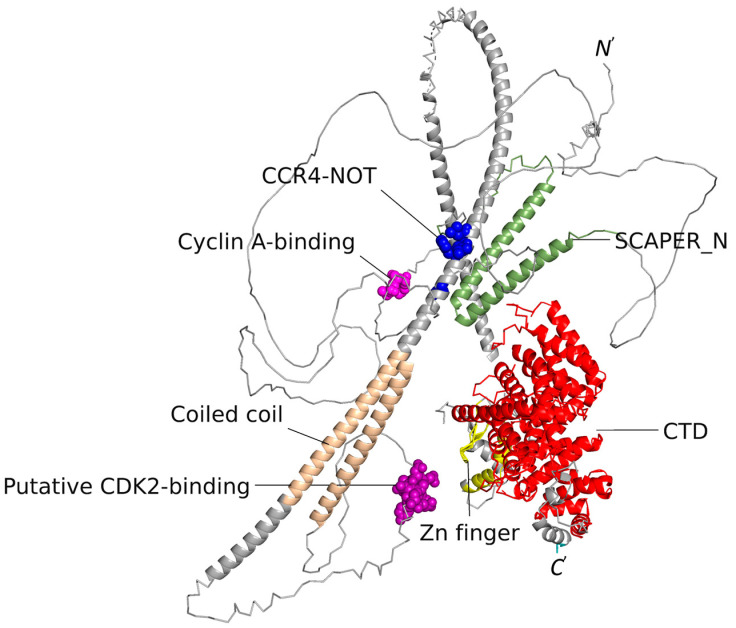
**The AlphaFold2-predicted model of the SCAPER protein.** The parts predicted with a confidence of 70% or higher are shown as thick ribbons and the rest of the protein is shown as a thin line. The protein is colored in grey, with functional domains colored as follows: the SCAPER_N domain (positions 90–183) is colored in green, the *RSL* cyclin A-binding motif is in pink (199–201, shown as spheres), the zinc finger residues (792–812) are colored yellow, and the C-terminal domain (CTD, positions 859–1357) is colored in red. The protein also contains a coiled coil region (colored wheat), whose function is still unknown. E620, F628, I629, and L632, which were implicated by Höpfler and co-workers [25] in CCR4–NOT complex binding, are shown as blue spheres. Finally, a putative CDK2-binding region, predicted by us (see below), is shown as magenta spheres.

**Figure 2 genes-15-00791-f002:**
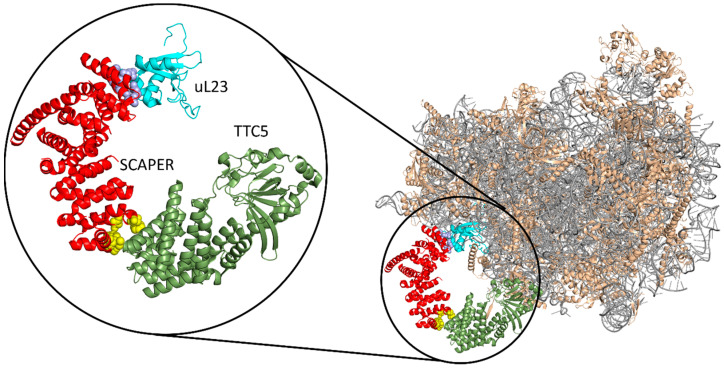
**Interactions of SCAPER’s CTD with the ribosome.** Right: the full structure. SCAPER’s CTD is colored red, TTC5 is colored olive green, the uL23 ribosomal protein is colored cyan, the other ribosomal proteins are colored wheat, and the ribosomal RNA is grey. Left: a blowup of the interaction. The SCAPER residues interacting with TTC5 and uL23 are shown as spheres, colored yellow and slate blue, respectively.

**Figure 3 genes-15-00791-f003:**
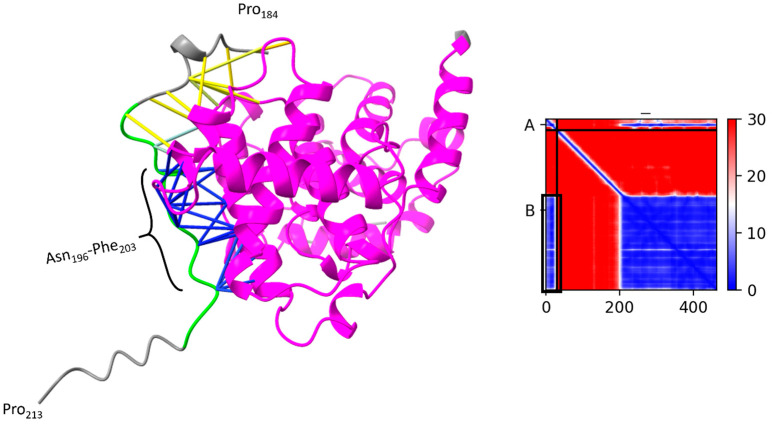
**SCAPER’s Cyclin A-binding region.** The image on the left shows the AlphaFold-Multimer-predicted complex between residues 184 and 213 of SCAPER (short ribbon) and residues 177 and 432 of human cyclin A2 (UniProt ID P20248, colored pink). The position of SCAPER’s residues 192–207 (green) with respect to cyclin A was predicted with high confidence, as indicated by the predicted aligned error (PEA) confidence metric, shown on the right (A, B denote the SCAPER and Cyclin A2 chains, respectively, and the part corresponding to residues 192–207 is marked by the black rectangle). Within this region, all SCAPER residues that are within 4 Å of cyclin A are marked by sticks. The blue sticks, which involve residues 196–203, denote the part of SCAPER that is both closely and confidently interacting with cyclin A.

**Figure 4 genes-15-00791-f004:**
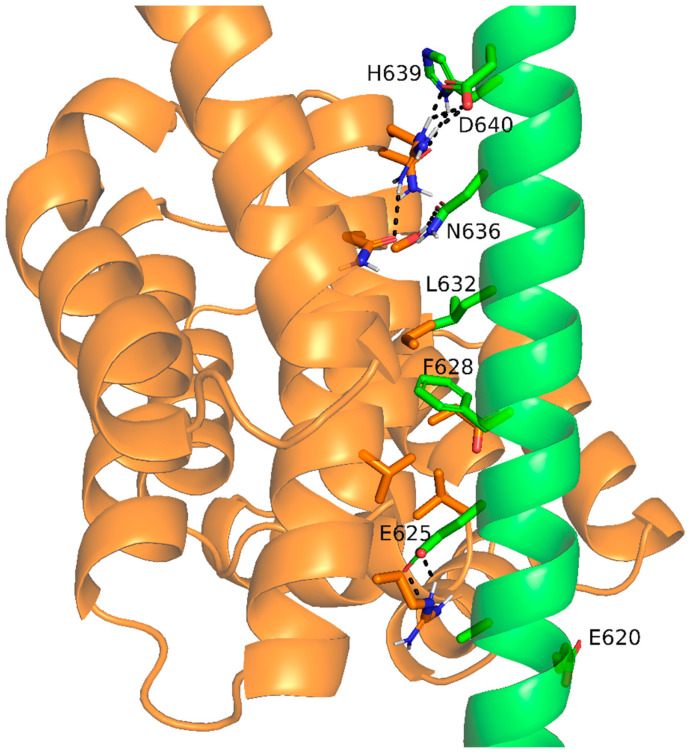
**AlphaFold-predicted interactions between SCAPER (green) and CNOT11 (orange).** The prediction was performed using SCAPER’s sequence, which corresponds to the long helix (positions 548–684) and the C-terminal domain of human CNOT11 (UniProt ID Q9UKZ1, positions 225–512). AlphaFold2-multimer was run with relaxation of the predicted model to optimize side chain conformations and their non-covalent interactions. SCAPER residues that form direct interactions with CNOT11 are shown as sticks and labeled. Polar interactions (hydrogen bonds and salt bridges) are shown as dashed black lines.

**Figure 5 genes-15-00791-f005:**
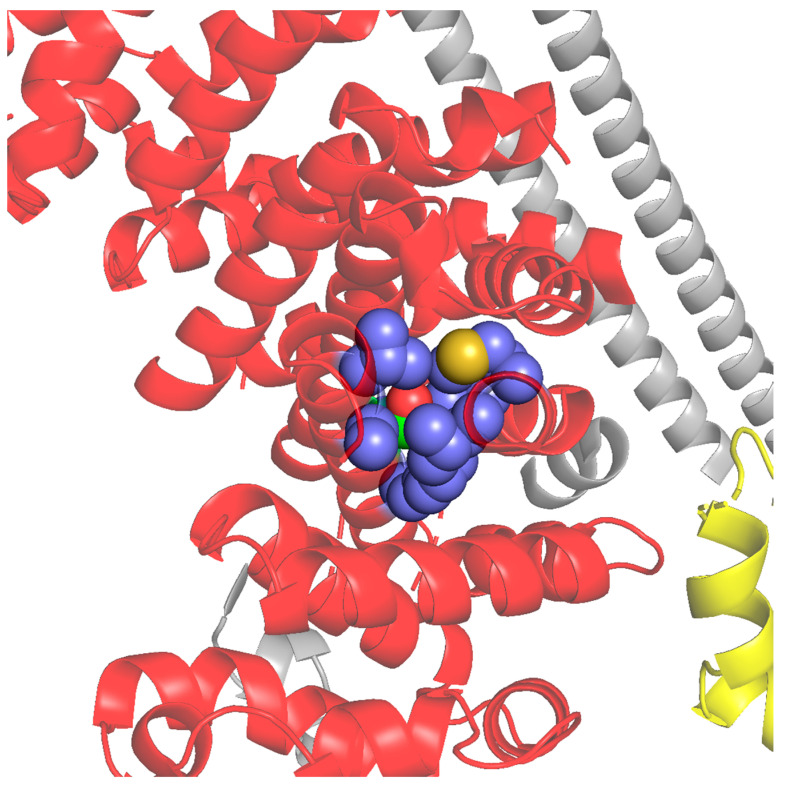
**Location and packing of S1219 in SCAPER’s CTD.** The CTD is presented as in Figure 1, except that it is rotated 90° to the left. S1219 (green) and the surrounding residues (blue) are shown as spheres, demonstrating their tight packing and the burial of S1219 inside the core of the CTD.

**Table 1 genes-15-00791-t001:** Relative frequencies of various clinical features for *SCAPER* gene reported cases.

Variable	Relative Frequency (N = 37)
Total number of reported patients	37
**Gender:**	
Male	14
Female	19
Not reported	4
**Consanguinity**	26/29 ^a^ (~90%)
**BMI:**	
Underweight (<18.5)	4/19 (21%)
Normal (18.5–25)	5/19 (26.3%)
Overweight (25.1–25)	1/19 (5.3%)
Obese (>30)	9/19 (47.4%)
**ID**	36/37 (97%)
**Behavior Issues:**	
ADHD	13/25 (52%)
Self-Injury	2/15 (13.3%)
**Developmental delays:**	
Sitting delay	9/11 (~82%)
Standing delay	4/9 (44.5%)
Walking delay	17/23 (74%)
Speaking delay	18/19 (~95%)
**MRI neuroimaging (normal finding)**	16/18 (~89%)
**Skeletal abnormalities**	4/15 (~27%)
**Facial dysmorphism**	10/31 (~32%)
**Hypotonia**	4/15 (~27%)
**EYE:**	
RP	31/33 (94%)
Strabismus	10/18 (56%)
Myopia	10/18 (56%)
Vision problems	23/30 (77%)
Cataract	7/19 (37%)
Nystagmus	2/14 (14%)
Glaucoma	2/3 (67%)
Elevator palsy	1/3 (33%)

^a^: When a particular clinical feature is not mentioned in the article for that specific patient, then that patient is excluded from the rest of the patients; therefore, the total number of patients will be reduced.

**Table 2 genes-15-00791-t002:** *SCAPER* variants identified in patients with RP and/or ID.

#	Nucleotide Change ^1^	Amino Acid Change ^1^	Main Phenotype	Mutant Allele Frequency (gnomAD)	Classification	References
1	c.352_354	(Y118fs*)	ID	UR	P	[2]
2	c.2973_2976del	p.I991MfsX26	RP, ID	UR	P	[5]
3	c.1859_1861del c.3656G>A	p.E620del p.S1219N	RP, ID	UR 1.86 × 10^−6^	P P	[5]
4	c.2023-2A>G	NA	RP, ID	UR	P	[5,8], Current cases
5	c.2179C>T c.1116delT	p.R727* p.V373Sfs*21	RP, ID	1.93 × 10^−6^ 6.85 × 10^−7^	LP LP	[3]
6	c.358C>T	p.R120*	ID	UR	P	[4]
7	c.2806delC	p.L936*	RP, ID	UR	P	[6]
8	c.2236dupA	p.I746Nfs*6	RP, ID	1.43 × 10^−6^	P	[9]
9	c.829C>T c.3707_3708delCT	p.R277* p.S1236Yfs*28	RP, ID	1.37 × 10^−6^ 6.58 × 10^−6^	P P	[9]
10	c.1495+1G>A c.3224delC	NA p.P1075Qfs*11	RP, ID	1.25 × 10^−6^ 2.05 × 10^−6^	P P	[9]
11	c.2377C>T c.2166-3C>G	p.Q793* NA	RP, ID	6.57 × 10^−6^ UR	P P	[9]
12	Chr15: 77,018,886–77,028,490	p.V623fs	RP, ID	UR	UR	[7]
13	c.1096C>T	p.R366*	RP, ID	9.92 × 10^−6^	P	[7]
14	c.1092dupT	p.V365Cfs*5	RP, ID	UR	LP	[7]
15	c.1883T>G	p.F628C	RP, ID	UR	UR	[7]
16	c.3781delG c.868_869delGA	p.V1261Sfs*26 p.E290Sfs*7	RP	UR 1.24 × 10^−6^	LP LP	[10]

Abbreviations: gnomAD: Genome Aggregation Database (v4.1.1); ID: intellectual disability; LP: likely pathogenic; NA: not available; P: pathogenic; RP: retinitis pigmentosa; UR: unreported. ^1^ Reference nucleotide and amino acid sequences are from NM_020843.4 and NP_065894.2.

**Table 3 genes-15-00791-t003:** Phenotypic characteristics of our patients affected with biallelic *SCAPER* variant (three patients).

Clinical Features	Patient A	Patient B	Patient C
Gender	F	F	F
Age (Years)	30	29	14
Height (cm)	147	153	152
Weight (Kg)	71.9	50	86
OFC (cm)	53.5	53.5	56
BMI	33.3	21.4	37.2
**Development:**			
Standing (m)	NA	NA	NA
Waking (m)	15	16	15
Speaking (m)	24	24	18
Sitting (m)	NA	NA	NA
ID (IQ)	52	55	59
Seizure	-	-	-
**Behavior Issues:**			
ADHD	+	+	+
Self-injury	-	-	-
Abnormal neuroimaging (MRI)	Nonspecific findings	NA	NA
Brachydactyly	+	+	+
**EYE:**			
RP	+	+	+
Strabismus	+	+	+
Myopia	+	-	-
Vision	+	+	+
Cataract	+	-	-
Nystagmus	+	+	-
Glaucoma	+	+	-
Elevator palsy	-	+	-
Facial appearance	Facial dysmorphism	Facial dysmorphism	Facial dysmorphism
Skeletal Abnormalities	Short stature	Short stature	Short stature
Speech	Monotonic nasal	Monotonic nasal	Monotonic nasal
Hypotonia	++	++	++

F: Female, NA: Not available, -: Negative, +: Positive, ++: Highly positive.

## Data Availability

No new data were created or analyzed in this study.

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
