# Peer review of "SCAPER-Related Autosomal Recessive Retinitis Pigmentosa with Intellectual Disability: Confirming and Extending the Phenotypic Spectrum and Bioinformatics Analyses"

_genes, 2024, doi:10.3390/genes15060791_

Round 1

Reviewer 1 Report

Comments and Suggestions for Authors

In their latest article, Sharkia et al. explore mutations in the SCAPER gene, previously linked to retinitis pigmentosa and intellectual disability. They describe three patients from an Arab consanguineous family in Israel with similar SCAPER syndrome features and additional ocular symptoms like nystagmus, glaucoma, and elevator palsy. The study includes a comprehensive analysis of all 37 cases described in the literature, confirming and expanding the clinical manifestations of SCAPER-related disorders.

The introduction is very informative and concise. It covers all related cases in chronological order and explains their relevance in an interesting way.

The provided tables are also very informative and helpful in recognizing the clinical features, rarity, and diversity of SCAPER variants.

The figures and figure legends are very high quality and clear.

Their study not only adds to the phenotypic spectrum of SCAPER variants but also to the interactions of the SCAPER protein with the help of machine learning.

479-481 The sentence doesn't seem to be finished.

All in all, a human geneticist with a patient with a SCAPER variant could find all relevant information on the topic in this article.

Comments on the Quality of English Language

It is very well-written and easy to follow. The use of the English language is mostly appropriate, with some exceptions. For example, the sentence " it is not clear that the causes of this symptom is of behavioral or metabolic background." (430-431) is unclear. 

Author Response

Reviewer 1:

We thank you for your positive and encouraging comments, we also thank you for pointing out the incomplete sentence as well as the clarification of the mentioned statement. The changes have been highlighted with yellow color. Please note that there are other changes and modifications suggested by the other reviewers.

Query: 479-481 The sentence doesn't seem to be finished.

Answer: the sentence in lines 479-481 has been completed.

Query: the sentence " it is not clear that the causes of this symptom is of behavioral or metabolic background." (430-431) is unclear.

Answer: the sentence has been modified so as to clarify the meaning.

Reviewer 2 Report

Comments and Suggestions for Authors

a.     Inconsistent use of tense and terminology.

b.     Sometimes "ID" is used for intellectual disability, and other times it's written out.

c.     Some sentences are lengthy and complex, making them difficult to read.

d.     Acronyms such as "RP" for retinitis pigmentosa are not always defined on first use.

e.     Line 43: "homozygous frameshift SCAPER variant as the cause of non-syndromic intellectual disability (ID) in a small Iranian family" - The term "small Iranian family" could be better phrased as "a family from Iran."

f.      Line 52: "short stature, obesity, and brachydactyly." This list should be separated from the preceding sentence for clarity.

g.     Line 55: "non-syndromic arRP" should be defined (non-syndromic autosomal recessive Retinitis Pigmentosa).

h.     Line 87: "using standard procedures" is vague and could be more specific.

i.      Line 91: The notation "NC_000015.10 (NM_020843.4)" should be clearly explained for readers not familiar with genetic notation.

j.      Lines 128-129, 144-145, 155-156: Use consistent language for pregnancy and delivery descriptions.

k.     Line 134: "with IQ of 52" should be "with an IQ of 52".

l.      Line 137: "She undergone a strabismus repair surgery" should be "She underwent strabismus repair surgery."

m.   References should be checked for consistency and formatting. Ensure that all references are complete and correctly cited.

n.     Line 138: Height and weight should include units (e.g., cm, kg).

o.     Line 170-172: The percentages for behavioral disorders and their occurrences should be clearly presented.

Comments on the Quality of English Language

Minor editing of english language needed

Reviewer 3 Report

Comments and Suggestions for Authors

The study is well designed, and the manuscript is well written with experimental evidence. The authors did an interesting study here and presented in a good manner. 

Author Response

Reviewer 3:

We thank you very much for your positive and encouraging evaluation. Kindly, note that there are changes and modifications suggested by the other reviewers. The changes have been highlighted with yellow color.